# Transformer-Based *HER2* Scoring in Breast Cancer: Comparative Performance of a Foundation and a Lightweight Model

**DOI:** 10.3390/diagnostics15172131

**Published:** 2025-08-23

**Authors:** Yeh-Han Wang, Min-Hsiang Chang, Hsin-Hsiu Tsai, Chun-Jui Chien, Jian-Chiao Wang

**Affiliations:** 1Department of Anatomical Pathology, Fu Jen Catholic University Hospital, Fu Jen Catholic University, New Taipei City 24352, Taiwan; yehanwang@gmail.com; 2School of Medicine, College of Medicine, Fu Jen Catholic University, New Taipei City 24205, Taiwan; 3Li Jen Pathology Clinic, Taipei 11469, Taiwan; 4AI Lab., Quanta Computer Inc., Taoyuan City 33377, Taiwan; david_tsai@quantatw.com (H.-H.T.); hung-chia.chen@quantatw.com (C.-J.C.); joe_wang@quantatw.com (J.-C.W.)

**Keywords:** artificial intelligence, breast neoplasms, epidermal growth factor, immunohistochemistry, in situ hybridization, observer variation

## Abstract

**Background/Objectives**: Human epidermal growth factor 2 (*HER2*) scoring is critical for modern breast cancer therapies, especially with emerging indications of antibody–drug conjugates for *HER2*-low tumors. However, inter-observer agreement remains limited in borderline cases. Automatic artificial intelligence-based scoring has the potential to improve diagnostic consistency and scalability. This study aimed to develop two transformer-based models for *HER2* scoring of breast cancer whole-slide images (WSIs) and compare their performance. **Methods**: We adapted a large-scale foundation model (Virchow) and a lightweight model (TinyViT). Both were trained using patch-level annotations and integrated into a WSI scoring pipeline. Performance was evaluated on a clinical test set (*n* = 66), including clinical decision tasks and inference efficiency. **Results**: Both models achieved substantial agreement with pathologist reports (linear weighted kappa: 0.860 for Virchow, 0.825 for TinyViT). Virchow showed slightly higher WSI-level accuracy than TinyViT, whereas TinyViT reduced inference times by 60%. In three binary clinical tasks, both models demonstrated a diagnostic performance comparable to pathologists, particularly in identifying *HER2*-low tumors for antibody–drug conjugate (ADC) therapy. A continuous scoring framework demonstrated a strong correlation between the two models (Pearson’s r = 0.995) and aligned with human assessments. **Conclusions**: Both transformer-based artificial intelligence models achieved human-level accuracy for automated *HER2* scoring with interpretable outputs. While the foundation model offers marginally higher accuracy, the lightweight model provides practical advantages for clinical deployment. In addition, continuous scoring may provide a more granular *HER2* quantification, especially in borderline cases. This could support a new interpretive paradigm for *HER2* assessment aligned with the evolving indications of ADC.

## 1. Introduction

Breast cancer is one of the most common malignancies worldwide and remains a leading cause of cancer-related death among women. Human epidermal growth factor 2 (*HER2*) immunohistochemistry (IHC) scoring plays a pivotal role in guiding targeted therapy decisions. According to the ASCO/CAP guidelines, patients with *HER2*-positive tumors (3+) are eligible for anti-*HER2* therapies, such as trastuzmab, whereas equivocal cases (2+) require confirmatory in situ hybridization (ISH) testing to assess *HER2* gene amplification. In recent years, the development of novel anti-*HER2* agents, particularly antibody–drug conjugates (ADCs), has expanded treatment options. Clinical trials have demonstrated that ADCs significantly prolong progression-free and overall survival in patients with *HER2*-low tumors (1+ or IHC 2+ without amplification), which account for approximately 45–64% of all breast cancers [1,2,3]. Moreover, ADCs are now being explored in tumors with ultralow *HER2* expression (defined as 0 < *HER2* < 1+) [4,5]. However, the expanding therapeutic landscape has also intensified a longstanding challenge: the inherent subjectivity and limited consistency of *HER2* scoring among pathologists, especially in borderline categories.

*HER2* scoring is influenced by multiple factors, including fixation time, antibody clones, staining protocols, tumor heterogeneity, and most importantly, inter-observer variability [2,6,7]. When using standardized specimens, identical tissue sections, or whole-slide images (WSIs), inter-observer agreement for *HER2* IHC scoring can reach 86–92%, with Cohen’s kappa values between 0.7 and 0.8 [8]. However, most validation studies and international proficiency assessments were primarily designed to evaluate *HER2*-positive (3+) detection. Recent investigations focusing on *HER2*-low diagnoses have revealed greater variability in distinguishing *HER2*-low from *HER2*-negative, with agreement rates ranging from 33% to 80% across studies [4,8,9,10,11]. Due to these discrepancies, reader retraining was often required to achieve diagnostic alignment. Furthermore, in the context of *HER2*-ultralow detection, one study reported low binary consistency between 0 and ultralow classifications, with a Fleiss kappa of just 0.292 [12]. These variations highlight the urgent need for more reproducible scoring methods, as inconsistent classification results may propagate into both treatment decisions and the design of ADC-based studies. In this context, artificial intelligence (AI)-based approaches may offer complementary support for improving the consistency and efficiency of *HER2* scoring.

Although the first FDA-cleared image analysis algorithm for *HER2* scoring was launched in 2009, automatic *HER2* assessment remains one of the most actively explored topics in digital pathology. Over the past decade, advances in deep learning have markedly improved the precision and robustness of image analysis models. These approaches have employed various inference levels—including cell-, area-, patch-, and WSI-based designs—and typically generate quantitative results, along with heatmaps to aid visual verification [6,13,14,15,16,17,18]. Since the introduction of the *HER2*-low classification, several studies have demonstrated that AI-assisted scoring can enhance intra-observer consistency, particularly in borderline cases [7,19,20,21,22]. By quantifying the proportion of cells with different staining intensities, AI tools can help pathologists achieve more reproducible evaluations and may also support junior readers in aligning with expert-level interpretations. However, most prior approaches have adopted convolutional neural networks as the primary architecture, whereas more recent transformer-based models are designed to incorporate adjacent contextual information, mirroring how pathologists integrate surrounding features into diagnostic decisions. Recent advances in lightweight transformers (TinyVit) have demonstrated competitive performance in general image classification benchmarks, suggesting their potential for efficient deployment in specialized domains [23].

Pathology foundation models utilize large-scale, transformer-based AI architectures that are pretrained on vast WSI datasets across multiple organ systems [24,25,26,27]. The first published example, CTransPath, was introduced in 2022 and was trained on publicly available datasets, such as The Cancer Genome Atlas (TCGA) and the Pathology AI Platform (PAIP). In 2023, Virchow was released, a vision transformer-based model with 632 million parameters, pretrained on approximately 1.5 million WSIs from 119,629 patients at Memorial Sloan Kettering Cancer Center [26]. By leveraging exposure to diverse morphologic patterns, foundation models have demonstrated stronger generalization capabilities and superior performance in zero-shot or few-shot tasks than conventional neural networks. Also, these foundation models have outperformed traditional conventional neural network approaches across multiple tasks [24,25,26,27]. These studies indicated that transformer-based architectures can achieve better generalization and contextual understanding than convolutional approaches in histopathology applications. However, these performance gains come at the cost of substantial computational requirements. The size and complexity of foundation models pose practical limitations for clinical deployment, where inference speed, hardware efficiency, and scalability are critical considerations. As a result, compact transformer models with fewer parameters, which require less computing power, may provide a more feasible alternative for routine diagnostic workflows.

Therefore, we adapted two transformer-based models for automated *HER2* scoring in breast-cancer WSIs. One is a state-of-the-art large-scale foundation model (Virchow), representing the upper bound of current AI performance with extensive pretraining. The other is a lightweight transformer (TinyViT) with 21 million parameters, offering a computationally efficient alternative that may be more practical for on-premises deployment in clinical pathology settings. Their performance was evaluated across multiple levels, including patch-level classification, WSI-level scoring, and inference efficiency. In addition, we explored a continuous scoring approach to complement conventional *HER2* categorization and assess its potential interpretive value.

## 2. Material and Methods

### 2.1. Case Collection

In this study, 108 breast cancer *HER2* slides were included. Paraffin blocks were retrieved from the Li Jen Pathology Clinic between June 2022 and December 2023 (Table 1). Of the included slides, 25 (23%) were *HER2* 0, 36 (33%) were *HER2* 1+, 29 (27%) were *HER2* 2+, and 18 (17%) were HER 3+ according to the original pathology reports. Forty-nine slides were used to build the model at the patch level, and 66 slides (including 7 test slides also used for the patch classifier task) were used as a test set to evaluate WSI-level scoring. This study received approval from the Research Ethics Review Committee of the Far Eastern Memorial Hospital (No. 113141-E, date of approval: 5 June 2024). All blocks used in the clinical diagnosis setting were retrieved from the repository. These slides and images were anonymized and did not contain any personal information. Therefore, the requirement for informed consent was waived.

### 2.2. Specimen Staining and Image Acquisition

The retrieved paraffin-embedded tissue blocks had been fixed in 10% neutral formalin for 6–72 h before further processing. The blocks were cut into 5 μm thick sections and mounted on hydrophilic slides. The slides were stained with ready-to-use anti-*HER2* antibodies (4B5, Ventana Medical Systems, Oro Valley, AZ, USA) using a standardized Roche protocol on a Benchmark GX (Ventana Medical Systems), ensuring consistent staining conditions across all specimens. Then, the slides were scanned with a Hamamatsu S210 (Hamamatsu, Iwata City, Shizuoka, Japan) at 40× magnification (0.23 μm per pixel resolution), and these images were saved in NDPI format.

### 2.3. Data Preparation and Model Training

To quantify *HER2*’s intensity and visualize distributions as heatmaps, the AI workflow was based on an image patch classifier (Figure 1) [6]. First, a pathologist (Dr. A) annotated invasive carcinomas in WSIs, and the tumor areas were partitioned into image patches (256 × 256 pixels, equivalent to 60 μm × 60 μm, containing 10 to 30 cells). Then, Dr. A labeled 5109 patches (Table 2) into five categories: non-tumor, 0 (no staining), 1+ (more than 10% of cells with weak staining), 2+ (more than 10% of cells with moderate incomplete membranous staining), and 3+ (more than 10% of cells with strong complete membranous staining). During patch-level annotation, any patches containing staining artifacts or technical irregularities were systematically labeled as ‘non-tumor’ to strengthen the model’s robustness against similar technical variations. After a washout period of 8 weeks, Dr. A blindly relabeled 1966 patches without any previous labeling information to evaluate intra-observer variation, and the other pathologist (Dr. B) labeled a portion of the data (1200 patches) to evaluate inter-observer variation. This assessment was used to determine a reasonable endpoint for the algorithm training, because it was implausible for models to outperform intra-observer consistency.

Two transformer-based neural networks were adopted in this study (Appendix A). The first was Virchow, a state-of-the-art pathology foundation model with 632 million parameters based on the Vision Transformer architecture [26]. The second was TinyViT, a lightweight vision transformer model comprising 21 million parameters [27]. Both networks were trained on a dataset of image patches labeled by Dr. A to build *HER2* patch classifiers, referred to as Model V (Virchow) and Model T (TinyViT). No architectural modifications were made, nor were layers frozen to the base models.

After reaching the designated training endpoints, the models were integrated into a WSI *HER2* scoring workflow. Tumor regions within each *HER2*-stained WSI were first segmented by a model, which had been fine-tuned from our previously developed virtual cytokeratin algorithm [28]. This cytokeratin-supervised model predicted breast epithelial components, especially tumor parts, in IHC stains. By constraining patch classifiers to tumor areas, scoring accuracy improved, while computational load decreased, and artifact robustness was enhanced by excluding most non-tissue processing artifacts. Upon completion of inference, the system outputs the proportion of patches in each *HER2* intensity level (P_0_ to P_3+_), the final WSI-level *HER2* score, and a visual heatmap overlay for interpretability. According to the ASCO/CAP guidelines, *HER2* scoring is based on the percentage of tumor cells with specific membranous staining intensities. In our workflow, the proportion of patches classified into each intensity category was used as a proxy for the percentage of stained tumor cells, following the same approach that leverages patch-level predictions for WSI-level scoring. Specifically, our approach is analogous to that of Pham et al. [6], who employed a weakly supervised deep learning framework to create an interpretable *HER2* scoring model by aligning its outputs with ASCO/CAP clinical guidelines. In this phase, 66 WSIs were used to evaluate accuracy and consistency with the ground truth from pathology reports. Since the workflow quantified the area of each staining intensity, we also calculated a continuous *HER2* score (c-score = 0 × P_0_ + 1 × P_1+_ + 2 × P_2+_ + 3 × P_3+_) by weighted average of patch-level classification. Regarding computing power demand by both models, a computer with RTX4060 Ti GPU (16GB) was used to compare the inference times for five WSIs between the models.

### 2.4. Statistical Analysis

Python software (ver. 3.8.13; https://www.python.org/) and SPSS version 21 for Windows (Chicago, Armonk, NY, USA) were used to compute the performance metrics of the two models. The metrics included accuracy, recall, precision, and F1 score in patch-level classifications. Consistency between models and pathologists was evaluated by calculating the linear weighted kappa (LWK). Performance in WSI *HER2* scoring was also assessed by accuracy and LWK. The c-score correlation between the two models was calculated using Pearson’s correlation coefficient (r), and the inference time comparison was analyzed using the paired *t*-test. Statistical significance was set at *p* < 0.05.

## 3. Results

### 3.1. Intra- and Inter-Observer Consistency at the Patch Level

Dr. A relabeled approximately 40% of the dataset (*n* = 1966) after a washout period of 8 weeks. The results showed an accuracy of 76.2% and good consistency (LWK = 0.825) between the two sets of intra-observer labels (Figure 2A). The most disparate classes were non-tumor and *HER2* 1+. If non-tumor patches (*n* = 1200) were excluded, the accuracy increased to 85.6%, and the LWK value rose to 0.879. Inter-observer agreement between the two pathologists (Dr. A vs. Dr. B) showed an overall accuracy of 58.5% and an LWK value of 0.611. After excluding non-tumor regions, accuracy improved to 61.9% and LWK to 0.671, consistent with prior reports of variability in *HER2* IHC interpretation [6]. Therefore, we determined an accuracy of 70% or an LWK of 0.7 as our training endpoint.

### 3.2. Performance of the Two Models at the Patch Level

To evaluate the training stability and optimization behavior of the two backbone models, we compared their validation F1 scores and accuracy curves throughout training (Appendix A). Model V achieved a higher performance in earlier epochs, with a peak F1 score (0.771) at epoch 13 and the highest validation accuracy (79.4%) at epoch 5, followed by mild fluctuations throughout the training process. In contrast, Model T demonstrated a slow, gradual, but stable improvement over the training epochs, eventually reaching its peak F1 score (0.755) and validation accuracy (76.1%) at epoch 40.

In the test set, Model V reached an overall accuracy of 73.3% and an LWK of 0.738 (Figure 2C). Accuracy and LWK increased to 79.6% and 0.844 by excluding non-tumor patches, as Model V had difficulties in non-tumor patches and misrecognized some patches as 0 or 1+. Moreover, it tended to upgrade 1+ to 2+ and 2+ to 3+. Model V’s misclassification patterns can be attributed to its higher sensitivity to color intensity variations. This large foundation model appears to prioritize staining intensity over morphological context, leading to misidentification of non-tumor regions with background staining as weak positive (0 or 1+) patches and systematic upgrading of patches based primarily on color intensity. Although Model T had fewer parameters, it outperformed Model V and reached an overall accuracy of 76.9% and an LWK of 0.787 (Figure 2D). Without non-tumor patches, accuracy and LWK improved to 81.4% and 0.857, respectively. Model T also tended to upgrade patches, but it performed more effectively in non-tumor regions. Like for human raters, 1+ and 2+ patches were most difficult to assess for both models.

Overfitting assessment based on training dynamics (Appendix A) showed stable validation performance without significant degradation relative to training accuracy. Both models demonstrated appropriate convergence patterns: Model V reached a plateau early with mild fluctuations, while Model T showed gradual, stable improvement. Validation curves indicated generalization without excessive training data memorization. Furthermore, the final patch-level performance’s positioning between inter-observer (LWK = 0.611) and intra-observer agreement (LWK = 0.825) suggests realistic learning within human performance boundaries.

### 3.3. Performance of the Two Models at the WSI Level

In WSI *HER2* scoring (Figure 3A,B), Model V achieved a higher accuracy (83.3%) than Model T (78.8%). Regarding agreement with ground truth, Model V also performed better (LWK = 0.860) than Model T (LWK = 0.825). However, the difference in performance was not significant (*p* = 0.321), and the assessments by both models were equivalent to those by pathologists [9,22]. Model V and Model T misclassified 11 and 14 cases, respectively, sharing errors on eight slides (5 cases were 2+, 2 were 1+, and 1 was 0). All misclassified cases were off by one grade from the ground truth. Model T exhibited a balanced distribution of over- and underestimation, but Model V tended to underestimate (score 2+ to 1+), because more non-tumor patches were classified as 0 or 1+ (Figure 3C).

To assess the performance for clinical decision-making, three clinically relevant binary classification tasks were evaluated (Table 3): (1) identifying *HER2* IHC 3+ cases eligible for anti-*HER2* therapy, (2) stratifying 2+ cases requiring confirmatory ISH, and (3) differentiating *HER2*-low (1+/2+) from *HER2*-negative (0) cases. In the first binary setting (3+ vs. 0/1+/2+), Model V showed perfect accuracy (100%), whereas Model T achieved an accuracy of 95.5% with a sensitivity of 100%. After excluding 3+ cases, the second binary task evaluated the ability to distinguish 2+ from 0/1+. Model V reached an accuracy of 84.6% and a sensitivity of 55.6%, with eight missed cases. Model T achieved an accuracy of 80.8% and the same sensitivity of 55.6%, with two false positives and eight false negatives. In the last task for *HER2*-low identification (1+/2+ vs. 0), Model V, with an accuracy of 94.2% and a sensitivity of 94.6%, outperformed Model T, which only achieved an accuracy of 86.5% and a sensitivity of 86.5%. Both models were comparable to pathologists and state-of-the-art algorithms in other studies [6,8,9,12,20].

In terms of continuous *HER2* score (c-score), the two models strongly correlated (Pearson’s r = 0.995) across all 66 test cases (Figure 4). This suggests that they were highly consistent in interpreting the overall *HER2* staining intensity at the WSI level, although their tiered *HER2* scores might differ. Furthermore, their c-scores were visually associated with ground-truth *HER2* scores.

### 3.4. Inference Time for the Two Models

Runtime analysis was conducted on five test cases to compare the inference efficiency between the two models. The average processing time for Model V was 1024.6 (688.7–1875.8) s per WSI. In contrast, Model T required on average only 368.7 (292.4–438.4) s, resulting in a time reduction of 60.7% (52.0–77.2%, *p* = 0.013). These results demonstrated that Model T saved substantial computing power and inference time.

## 4. Discussion

In this study, we trained two deep learning structures, a large foundation model (Virchow, Model V) and a lightweight transformer (TinyViT, Model T), for automatic *HER2* scoring in breast cancer WSIs. Both models demonstrated high accuracy and substantial agreement with the pathologist-annotated ground truth in patch-level classification, WSI-level scoring, and binary decision categorization. While Model V showed slightly higher performance in WSI-level accuracy and agreement metrics, Model T achieved equivalent abilities with a significantly faster inference time. In addition, we introduced a continuous score framework for the two models to assess *HER2* expression, and their results were highly consistent. These findings highlight the practical feasibility of both models in *HER2* assessment and set the stage for deeper comparisons in learning dynamics, predictive behavior, and clinical utility.

This work was inspired by a previous study conducted by Pham et al., who demonstrated that patch-based *HER2* evaluation can achieve high concordance with cell-based ground truth in digital pathology workflows [6]. We adopted patches instead of cells as a proxy to evaluate staining intensity and completeness. This eased the manual annotation task and allowed us to evaluate intra-observer consistency, inter-observer consistency, and model performance at a fundamental level. Consistency among pathologists is limited, especially in interpreting weak-to-moderate staining (1+ and 2+). Even for the same pathologists, overall accuracy was 76.2%, and accuracy for 1+ patches was only 62.7% in the relabeling task, which provided benchmarks to assess the models’ performance. During training, the large-scale foundation model, Virchow, learned quickly and reached a plateau in early epochs, corresponding to the characteristics of foundation models. By contrast, the lightweight Model T grew gradually but eventually achieved a performance equivalent to that of Model V. This pattern is consistent with the expectation that foundation models may generalize quickly in early stages, but lightweight models can also effectively adapt provided sufficient supervised training.

In patch-level classifications, both models exhibited intra-observer-level performance. The lightweight Model T achieved slightly higher overall accuracy than Model V, indicating better alignment with the manually labeled ground truth. However, like for humans, 1+ and 2+ patches were the most difficult classification items for both models, and both models tended to upgrade patches. After reviewing these misclassified patches, we identified several plausible causes (see Appendix A for examples). First, many cases were borderline or equivocal even to pathologists, particularly when distinguishing between 1+ and 2+ cases. Second, strong cytoplasmic staining, often seen in 1+ patches, may have misled models to overestimate their intensity. Third, some 2+ patches with strong staining but insufficient membranous completeness were incorrectly upgraded to 3+. These patterns indicate that both models were more sensitive to staining intensity than completeness, mirroring common interpretive challenges in manual *HER2* assessment.

In WSI-level *HER2* scoring, both models achieved performances similar to those of experienced pathologists and other state-of-the-art algorithms [6,19,20,29]. Model V demonstrated a marginally higher accuracy and LWK than Model T, although the difference was not significant. While both models exhibited a tendency to overestimate *HER2* intensity at the patch level, their predictions diverged when scores were aggregated at the WSI level: Model T demonstrated a more balanced distribution of over- and underestimation, whereas Model V tended to predict lower scores, particularly in 2+ cases. This discrepancy may be attributed to differences in the processing of non-tumor regions. Although a tumor masking algorithm was applied to restrict inference to tumor areas, benign glands and stromal components were sometimes inevitably included. Both classifiers occasionally misclassified these regions as 0 or 1+, thereby reducing the relative proportion of 2+ patches (Figure 3C). This effect was more pronounced in Model V, which showed lower accuracy (66.3%) in identifying non-tumor patches. As a result, Model V produced more conservative *HER2* scores than Model T at the WSI level, with clinical significance for borderline 2+ cases where small changes in patch proportion calculations can shift final classification below diagnostic thresholds.

To investigate clinical applicability, we further evaluated the models across three binary classification tasks that reflected common decision points in HER assessment. The first task focused on identifying HER 3+ cases eligible for anti-*HER2* therapy. Both models performed excellently, and Model V even achieved perfect accuracy. These results suggest that both models are highly reliable in recognizing *HER2* overexpression. The second task aimed to stratify 2+ cases that require confirmatory ISH testing. Both models reached moderate accuracy (~80–85%), but sensitivity was limited (55.6%), likely due to underestimation caused by misclassification of non-tumor regions as 0 or 1+. Several strategies could improve sensitivity for the identification of 2+ cases. For example, ensemble approaches combining both models could leverage their complementary strengths, as Model T demonstrated better non-tumor discrimination, while Model V showed superior overall accuracy. Additionally, confidence scoring mechanisms could flag borderline cases for manual review, particularly when patch-level predictions show high variance within WSIs. For such borderline cases, the manual review of *HER2* heatmaps and verification of 2+ patch proportions were still necessary, highlighting the role of human-in-the-loop decision-making. The third task differentiated *HER2*-low from *HER2*-negative cases to identify potential candidates for ADC therapy. Both models performed similarly to experienced pathologists and previously published AI approaches [4,8,9,10,12]. Model V achieved a higher accuracy and sensitivity (~94%), whereas Model T reached ~86% in both parameters. These results indicate the feasibility of automated *HER2* scoring as a decision-support tool, providing consistent, interpretable, and visually guided outputs for clinical integration. At the same time, these findings also reflect the inherent limitations of tiered scoring systems, particularly in borderline or heterogeneous cases [30,31].

The persistent challenges in 1+ and 2+ scoring reflect fundamental limitations at multiple analytical levels. At the patch level, 256 × 256 pixel patches face inherent difficulties in capturing membranous completeness and cellular proportion criteria that distinguish these borderline categories. Unlike 0 and 3+ cases with clear intensity differences, 1+ and 2+ assessments depend heavily on subtle morphological features that may be obscured at the current patch resolution. At the WSI level, borderline cases frequently contain heterogeneous patch populations with mixed staining patterns. When a WSI contains, for example, around 10% of 1+ or 2+ patches, the tier-based classification system forces a binary decision that may not reflect the underlying biological complexity. This aggregation challenge highlights the inherent limitations of discrete scoring systems for capturing the continuum of *HER2* expression. To better capture the continuum of *HER2* expression, we further explored a continuous scoring approach.

To provide a more nuanced quantification of *HER2* expression, we introduced a continuous scoring framework (c-score) based on the proportion and intensity of predicted patch classes [32]. Despite architectural differences, the c-scores generated by Model V and Model T were highly correlated, suggesting consistent interpretation of *HER2* expression at the WSI level. While their final categorical scores occasionally differed, the underlying c-scores reflected similar signal distributions, capturing the continuum of *HER2* expression more faithfully than discrete labels. Moreover, c-scores exhibited a clear visual association with pathologist-assigned *HER2* categories, offering an interpretable and intuitive measure that may aid in borderline or equivocal cases. This approach complements other strategies, such as the seven-class subdivision proposed by Pedraza et al. [33], which also aims to improve the handling of ambiguous cases by expanding the decision boundaries for the equivocal 2+ category before aggregating back to standard scores. Although further validation is needed, these findings highlight the potential of this c-score as a proof of concept for continuous *HER2* assessment, suggesting a future direction for AI-assisted precision pathology.

In addition to accuracy and interpretability, inference efficiency is a practical consideration for clinical deployment of AI models. Runtime analysis revealed a substantial difference between the two models: the lightweight Model T completed WSI-level inference in approximately one-third of the time required by Model V. This significant reduction in processing time decreased computational resource demands and enhanced scalability in high-throughput clinical workflows. While foundation models like Model V can offer strong generalization and rapid early-stage convergence, their computational demands may pose challenges for routine use, particularly in resource-constrained settings. In contrast, lightweight models, such as Model T, demonstrate not only a competitive diagnostic performance but also superior inference efficiency, potentially lowering the barrier for real-world implementation in pathology laboratories. In this study, both models can operate on the same RTX4060 Ti GPU configuration (approximate workstation cost: USD 5000–8000), making the hardware barrier relatively low for clinical deployment. However, for high-throughput laboratories requiring faster Model V processing, upgrading to enterprise-grade GPUs (e.g., A100) may be considered, with infrastructure costs reaching USD 15,000–20,000. For clinical integration, our models are designed for compatibility with existing digital pathology workflows. The most effective implementation would involve integration with hospital information systems or established digital pathology platforms, enabling seamless incorporation into routine diagnostic workflows. Additionally, considering open-source model deployment could facilitate broader adoption across different institutional platforms and promote collaborative development. The interpretable outputs, including quantitative patch percentages and visual heatmaps, are designed to complement rather than replace pathologist expertise, supporting evidence-based diagnostic decisions within existing clinical frameworks.

Another promising future direction is to explore the feasibility of predicting *HER2* status directly from H&E-stained images, a method proposed by researchers like Wang et al. [34]. Their model, named HAHNet, was designed to classify *HER2* status using H&E-stained histological images, thereby potentially reducing the additional costs associated with IHC staining and alleviating the burden on pathologists. The study reported that their HAHNet model achieved superior performance compared to other computational methods for H&E-based prediction. While our study and many others focus on IHC-stained images, this alternative approach represents a valuable avenue for streamlining the diagnostic workflow and is a significant area for future research.

This study has some limitations. First, although our test set was derived from real-world diagnostic cases, the total number of cases (*n* = 66) was relatively small, and 7 cases for patch classifier testing were reused in the WSI test set. To prevent data leakage, these overlapping cases were rigorously excluded from training and validation phases, ensuring independent evaluation at both analytical levels. Furthermore, all samples were sourced from a single institution using standardized protocols; thus, this represents an initial model development phase with controlled technical conditions. Multicenter validation datasets are essential to confirm the generalizability of the models’ performance, particularly across different staining protocols, scanner platforms, and *HER2* prevalence distributions. Future work will focus on multi-institutional studies to assess the models’ robustness across varied technical and demographic conditions for clinical deployment. Second, model training and evaluation relied on pathologist annotations as the reference standard. While intra- and inter-observer consistency were assessed, subjective variability in *HER2* scoring remains an inherent challenge, especially in borderline categories, such as 1+ and 2+. The accuracy of tumor masking algorithms may also influence models’ output, particularly in WSI-level scoring. Third, although the proposed c-score offers a potential measurement for the granular interpretation of *HER2*, its clinical utility is yet to be validated in prospective or outcome-linked studies. Whether this c-score can meaningfully inform treatment stratification, especially for *HER2*-low populations, will require further investigations in larger, annotated cohorts.

## 5. Conclusions

We developed two transformer-based models for automated *HER2* scoring in breast cancer WSIs and demonstrated high concordance with pathologist annotations across multiple tasks. While the foundation model achieved slightly higher accuracy, the lightweight model offered equivalent performance with faster inference. In addition, the proposed continuous c-score captured *HER2* expression as a spectrum, offering a promising framework for interpretable, quantitative assessment. These findings support the clinical applicability of AI-assisted *HER2* scoring and highlight opportunities for future precision pathology.

## Figures and Tables

**Figure 1 diagnostics-15-02131-f001:**
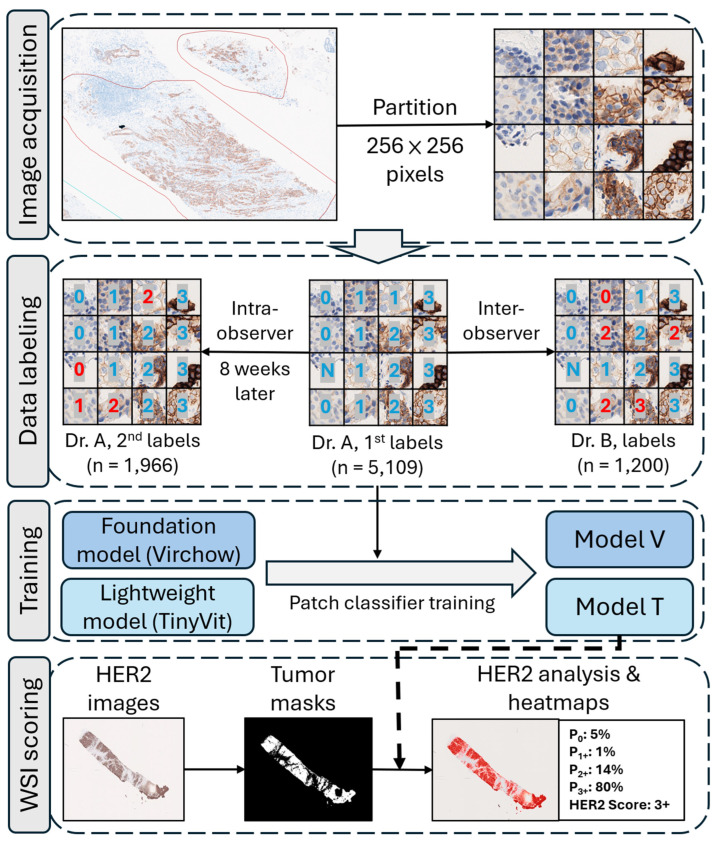
Overview of the *HER2* scoring workflow. Two transformer-based models (Virchow and TinyViT) were trained using pathologist-annotated patches to classify *HER2* immunoreactivity into five categories (0, 1+, 2+, 3+, and non-tumor. Red means different from previous labeling). After patch-level training, the models were applied to WSIs to generate *HER2* predictions and heatmaps of tumor regions. The WSI-level score was estimated based on the percentage of predicted patch classes, following ASCO/CAP guidelines. Abbreviations: *HER2* = human epidermal growth factor 2; WSI = whole-slide image.

**Figure 2 diagnostics-15-02131-f002:**
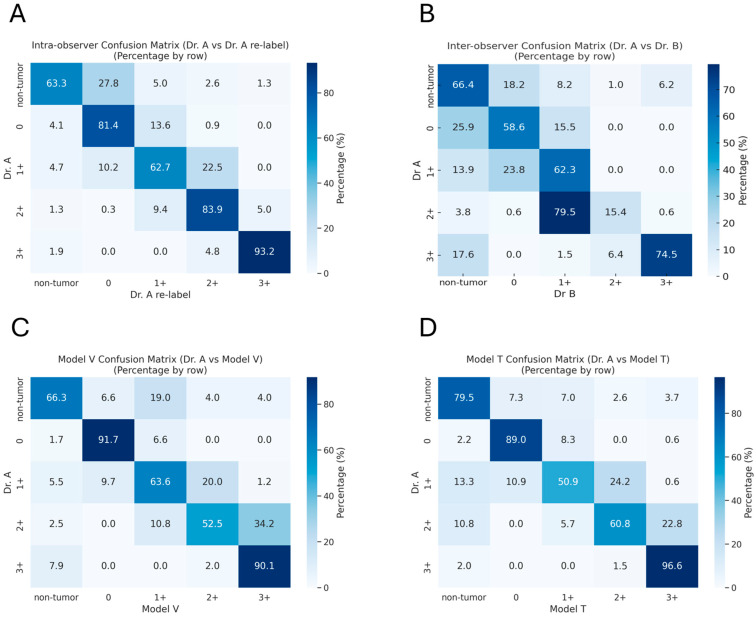
Confusion matrices for patch-level *HER2* classification. (**A**) Intra-observer comparison (Dr. A initial labeling vs. Dr. A relabeling). (**B**) Inter-observer comparison (Dr. A vs. Dr. B). (**C**) Dr. A vs. Model V. (**D**) Dr. A vs. Model T. Each matrix shows row-normalized percentages by ground truth category. Abbreviation: *HER2* = human epidermal growth factor 2.

**Figure 3 diagnostics-15-02131-f003:**
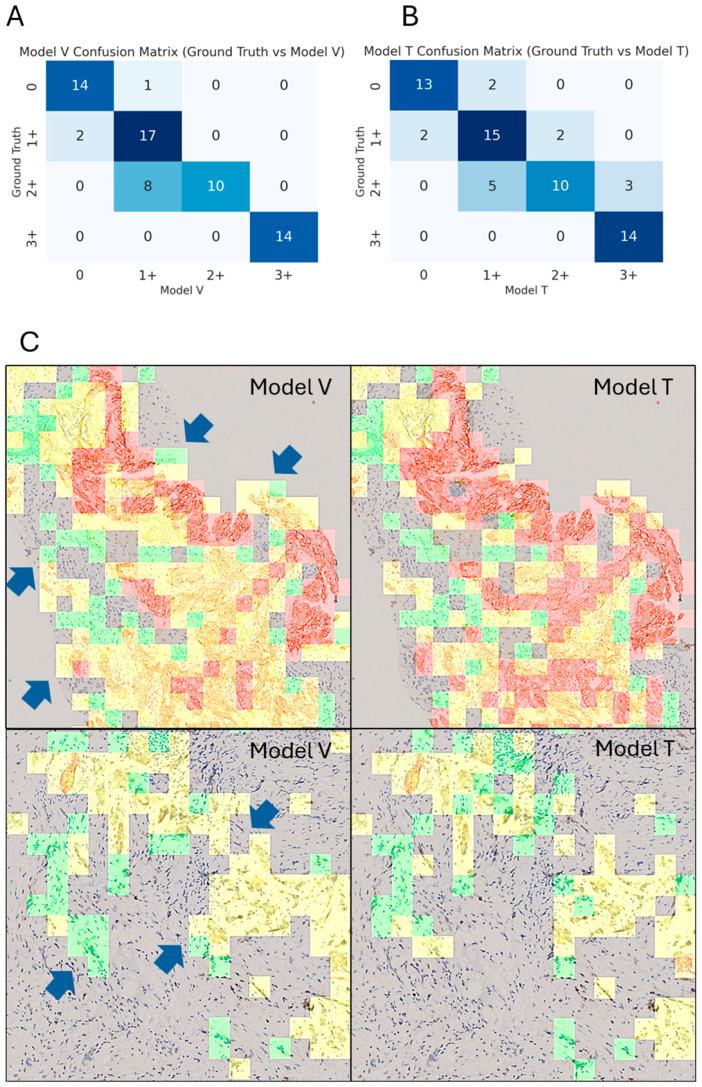
Confusion matrices and *HER2* heatmaps of the two models for WSI-level *HER2* scoring. (**A**) Model V vs. ground truth. (**B**) Model T vs. ground truth. Ground truth was according to original reports. (**C**) Heatmap examples of Model V (**left**) and Model T (**right**). Model V misclassified more of the non-tumor patches as 0 or 1+ (blue arrows). Abbreviations: *HER2* = human epidermal growth factor 2; WSI = whole-slide image.

**Figure 4 diagnostics-15-02131-f004:**
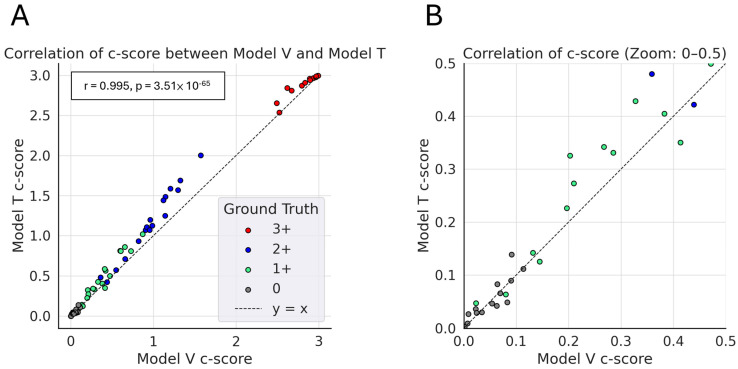
Correlation of c-scores between Model V and Model T. (**A**) Scatter plot showing the correlation between c-scores generated by Model V (*x*-axis) and Model T (*y*-axis) across all test cases (*n* = 66). Each dot represents a single case, with colors indicating *HER2* IHC ground truth: red for 3+, blue for 2+, light green for 1+, and gray for 0. (**B**) Zoomed-in view of the same scatter plot, focusing on the c-score range 0−0.5. Abbreviations: *HER2* = human epidermal growth factor 2; IHC = immunohistochemistry.

**Table 1 diagnostics-15-02131-t001:** Case details.

	*HER2*: 0	*HER2*: 1+	*HER2*: 2+	*HER2*: 3+	Total
Dataset	13	18	12	6	49
Training	5	13	10	3	31
Validation	5	4	1	1	11
Test	3	1	1	2	7
WSI test set	15	19	18	14	66 *

* Includes 7 WSIs in test set. To prevent data leakage, the 7 overlapping slides between patch and WSI test sets were completely excluded from all training and validation phases, serving solely as independent test cases for both patch-level and WSI-level evaluation. Abbreviations: *HER2* = human epidermal growth factor 2; WSI = whole-slide image.

**Table 2 diagnostics-15-02131-t002:** Number of patches for model training.

	Non-Tumor	0	1+	2+	3+	Total
Training	986	794	534	282	91	2687
Validation	541	245	175	158	323	1442
Test	273	181	165	158	203	980
Total	1800	1220	874	598	617	5109

**Table 3 diagnostics-15-02131-t003:** Model performance in three clinical binary tasks.

Binary Tasks	Model V (Virchow)				Model T (TinyVit)			
	Accuracy	Precision	Sensitivity	F1 Score	Accuracy	Precision	Sensitivity	F1 Score
0, 1+, 2+ vs. 3+ *	100%	100%	100%	1.00	95.5%	82.4%	100%	1.00
0, 1+ vs. 2+ **	84.6%	100%	55.6%	0.714	80.8%	83.3%	55.6%	0.667
0 vs. 1+, 2+ ***	94.2%	97.2%	94.6%	0.959	86.5%	94.1%	86.5%	0.901

* Includes all 66 test cases, corresponding to selecting patients eligible for anti-*HER2* therapy. ** Excludes 3+ cases, corresponding to select candidates for ISH test. *** Represents *HER2*-low vs. *HER2*-negative, based on the ASCO/CAP and emerging ADC treatment criteria. Abbreviations: ADC = antibody–drug conjugate; *HER2* = human epidermal growth factor 2; ISH = in situ hybridization.

## Data Availability

The original contributions presented in the study are included in the article; further inquiries can be directed to the corresponding author.

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
