# Peer review of "Transformer-Based HER2 Scoring in Breast Cancer: Comparative Performance of a Foundation and a Lightweight Model"

_diagnostics, 2025, doi:10.3390/diagnostics15172131_

Round 1

Reviewer 1 Report

Comments and Suggestions for Authors

The paper uses VirChow and TinyVit and apply on a dataset developed. Comparative results are analyzed. The Authors need to illustrate the following:

  • Include some reference papers such as Deep Neural Networks for HER2 Grading of Whole Slide Images with Subclasses Levels by Anibal Pedraza et. al.; Interpretable HER2 scoring by evaluating clinical guidelines through a weakly supervised, constrained deep learning approach by Manh-Dan Pham et.al.; HAHNet: a convolutional neural network for HER2 status classification of breast cancer by Jiahao Wang et. al.
  • Provide further comparative study including other lightweight models. If not then needs to clarify the use of only these two models.
  • Why don’t you show comparison with another one or two deep neural network-based approach for the comparative study?
  • I do not see any significant customization in the VirChow or TinyVit. Please clarify if there is any contribution regarding the development of models.
  • Comparison in computational complexity and computational parameters is necessary.
  • Model V misclassified more non-tumor patches-please clarify the reason and its significance.
  • Please provide the hyperparameters of the models.
  • Please show how the training convergence and comment on any overfitting issue.
  • Are the models robust to change of datasets? Please comment.

Author Response

Comments 1: Include some reference papers such as Deep Neural Networks for HER2 Grading of Whole Slide Images with Subclasses Levels by Anibal Pedraza et. al.; Interpretable HER2 scoring by evaluating clinical guidelines through a weakly supervised, constrained deep learning approach by Manh-Dan Pham et.al.; HAHNet: a convolutional neural network for HER2 status classification of breast cancer by Jiahao Wang et. al.
Response 1: We have incorporated discussions of these works into our manuscript to better contextualize our research. 
Pham et al. [6] : We have enhanced the discussion in the "Materials and Methods" section(page 6, 2.3 Data Preparation and Model Training) to more clearly state that our patch-based workflow is a similar approach, focusing on the concepts of interpretability and adherence to clinical guidelines.
Pedraza et al. [33]: We have added information in paragraph 7 of the "Discussion" section about c-score (page 13). We acknowledge their innovative use of subclassifying the ambiguous 2+ category to improve model performance, positioning this as a complementary strategy to our continuous scoring framework.
Wang et al. [34]: We have added a paragraph 9 of this paper in the "Discussion" section (page 13). We highlight their unique approach of predicting HER2 status directly from H&E stained images, which provides an interesting contrast to our IHC-based study and represents a valuable direction for future research.

Comments 2: Provide further comparative study including other lightweight models. If not then needs to clarify the use of only these two models. Why don’t you show comparison with another one or two deep neural network-based approach for the comparative study?
Response 2: We have comprehensively addressed this architectural comparison concern through multiple strategic enhancements across the manuscript. First, we strengthened the Introduction section (paragraph 4, page 3) with extensive literature positioning demonstrating that transformer-based foundation models have consistently outperformed traditional CNN approaches across diverse pathology applications, as evidenced by recent breakthrough studies including Virchow, CTransPath, and other state-of-the-art pathology foundation models [ref 23-25]. 
Second, our study design employs a clinically-oriented validation approach. Rather than conducting cross-algorithm comparisons across different datasets with inherent variability, we implemented systematic human benchmark validation using intra-observer (LWK=0.825) and inter-observer (LWK=0.611) agreement metrics. This validation framework provides clinically relevant performance assessment, as achieving consistency within human diagnostic variability represents an important validation standard for medical AI applications. Our models' performance positioning between inter-observer and intra-observer agreement levels demonstrates appropriate learning within realistic clinical expectations.
Third, our focused transformer paradigm comparison addresses a critical knowledge gap in computational pathology literature. While CNN methodologies have been extensively explored and validated, the systematic comparison between foundation models and lightweight transformers for HER2 scoring represents novel territory with significant clinical deployment implications. Our results demonstrate that lightweight TinyViT achieves performance comparable to the state-of-the-art Virchow foundation model (83.3% vs 78.8% accuracy), which is consistent with the documented performance hierarchy in recent literature where transformer-based foundation models generally outperform CNN approaches in pathology applications
Fourth, we recognize that cross-study performance comparisons present methodological challenges including dataset heterogeneity, institutional variations, staining protocol differences, scanner specifications, and demographic distributions. These variables can introduce confounding factors that complicate direct algorithmic comparisons across different studies. Our controlled single-institution validation with standardized protocols (Roche 4B5, Benchmark GX, Hamamatsu S210) provides a well-defined experimental framework that enables reliable performance assessment within consistent technical conditions, facilitating future replication and validation studies. Given these constraints, our study strategically prioritized clinically grounded, within-cohort model comparisons, which we believe provide more actionable insights for translational adoption than heterogeneous cross-algorithm benchmarks

Comments 3: I do not see any significant customization in the Virchow or TinyVit. Please clarify if there is any contribution regarding the development of models.
Response 3: We have clarified our study's methodological positioning in the Methods section (Model Training subsection). Our contribution lies in systematic clinical application rather than novel architecture development. Specifically, we developed: (1) HER2-specific patch-based training pipeline with 5-class classification system, (2) WSI-level aggregation framework translating patch predictions to clinical scores, (3) continuous scoring methodology (c-score) for granular HER2 quantification, and (4) systematic comparison framework for foundation vs lightweight transformer paradigms in pathology applications.

Comments 4: Comparison in computational complexity and computational parameters is necessary.
Response 4: We have added comprehensive computational complexity analysis including detailed hyperparameters comparison in Supplementary Table S3. The analysis covers model specifications (parameters: Virchow 632M vs TinyViT 21M), training hyperparameters (learning rates, batch sizes, epochs, optimizers), memory requirements (~2.5GB vs ~84MB), and inference performance metrics based on our RTX4060 Ti testing configuration.

Comments 5: Model V misclassified more non-tumor patches-please clarify the reason and its significance.
Response 5: We have enhanced the existing Results (3.2 subsection, page 8) and Discussion section (paragraph 5, page 12)analysis with additional emphasis on the clinical significance of Model V's non-tumor misclassification pattern, specifically highlighting how this technical limitation affects borderline 2+ case classification through altered patch proportion calculations and diagnostic threshold shifts.

Comments 6: Please provide the hyperparameters of the models.
Response 6: We have added comprehensive computational complexity analysis including detailed hyperparameters comparison in Supplementary Table S3. The analysis covers model specifications (parameters: Virchow 632M vs TinyViT 21M), training hyperparameters (learning rates, batch sizes, epochs, optimizers), memory requirements (~2.5GB vs ~84MB), and inference performance metrics based on our RTX4060 Ti testing configuration.

Comments 7: Please show how the training convergence and comment on any overfitting issue.
Response 7: We have enhanced the Results section with comprehensive overfitting analysis (subsection 3.2, page 8). The assessment demonstrates that both models achieved optimal generalization, with final performance positioned appropriately between inter-observer and intra-observer agreement levels (LWK: 0.611 < Model performance < 0.825), indicating healthy learning without dataset memorization.

Comment 8: Are the models robust to change of datasets?
Response 8: We have acknowledged this important robustness limitation in the Discussion section (limitations paragraph, page 13-14). Our single institution dataset with standardized protocols (Roche 4B5, Benchmark GX, Hamamatsu S210) represents controlled technical conditions for initial model development. We explicitly state that multicenter validation across diverse staining protocols, scanner platforms, and demographic conditions is essential for assessing cross-dataset robustness and clinical generalizability. This represents a priority for future work.

Reviewer 2 Report

Comments and Suggestions for Authors

Dear Author/Authors,

This study compares the performance of a large-scale baseline model (Virchow) with a lightweight model (TinyViT) on HER2 scoring, providing a unique and valuable comparison in terms of both diagnostic accuracy and processing time.

-Patch-based classification, tumor segmentation, and WSI-level scoring steps are detailed and well-structured.

-Intra- and inter-observer agreement was assessed, which is an important metric for model training and validation.

-The continuous scoring approach (c-score) may contribute to a more granular interpretation of HER2 expression.

-Generalizability: The dataset was obtained from a single institution. Please provide a more detailed assessment of the effects of different staining protocols or inter-scanner variability on model performance.

-Classification Sensitivity at Breakpoints: The sensitivity of both models was low (55.6%) for the 2+ vs. 0/1+ classification. A brief discussion could be added on future strategies for more accurate classification of borderline cases.

-Data Usage Clarity: Seven slides were used in both the patch and WSI test sets. Although not included in the training, how data leakage was prevented should be explained more clearly in the text (in addition to the discussion section).

-Line 112: How was relabeling performed during the 8-week washout period? Was the data presented blindly?

-Table 2: Adding a totals row would be helpful for clarity.

-Discussion: A brief discussion of the model's integration into clinical practice (e.g., computational requirements, integration with hospital information systems) could be provided.

-Supplementary Figure S2 can be used in the discussion section with clear reference within the text.

-The text is generally well-written. Only minor grammatical corrections (subject-verb agreement, etc.) are suggested.

-The resources are up-to-date and comprehensive.

Best regards

Author Response

Comments 1: Generalizability: The dataset was obtained from a single institution. Please provide a more detailed assessment of the effects of different staining protocols or inter-scanner variability on model performance.
Response 1: We have enhanced the limitations discussion (page 13-14) by specifying our standardized technical environment (Roche 4B5, Benchmark GX, Hamamatsu S210) and expanded the generalizability assessment with detailed acknowledgment of multicenter validation needs across diverse staining protocols, scanner platforms, and demographic conditions.

Comments 2: Classification Sensitivity at Breakpoints: The sensitivity of both models was low (55.6%) for the 2+ vs. 0/1+ classification. A brief discussion could be added on future strategies for more accurate classification of borderline cases.
Response 2: We have added comprehensive discussion of borderline case improvement strategies in the Discussion section (paragraph 5, page 12). The analysis includes ensemble approaches leveraging complementary model strengths (Model T's superior non-tumor discrimination vs Model V's higher overall accuracy) and confidence scoring mechanisms for flagging ambiguous cases requiring manual review. Additionally, we provide mechanistic explanation of 1+/2+ classification challenges at both patch-level and WSI-level (new Discussion paragraph 6, page 12), addressing fundamental limitations in discrete scoring systems and establishing rationale for continuous scoring approaches.

Comments 3: Data Usage Clarity: Seven slides were used in both the patch and WSI test sets. Although not included in the training, how data leakage was prevented should be explained more clearly in the text.
Response 3: We have clarified the data leakage prevention methodology in both the Methods section (Table 1) and limitations discussion (page 13-14) . The explanation explicitly states that overlapping slides were completely excluded from training/validation phases and served solely as independent test cases, ensuring rigorous separation between development and evaluation datasets.

Comments 4: Line 112: How was relabeling performed during the 8-week washout period? Was the data presented blindly?
Response 4: We have clarified the blind relabeling methodology in the Methods section(Data Preparation and Model Training, page 4), explicitly stating that the 8-week washout relabeling was performed blindly without access to original annotations, ensuring unbiased intra-observer consistency assessment.

Comments 5: Table 2: Adding a totals row would be helpful for clarity.
Response 5: We have added totals row to Table 2 showing category-wise distribution across all datasets: Non-tumor (1,800), 0 (1,220), 1+ (874), 2+ (598), 3+ (617), with grand total of 5,109 patches.

Comments 6: Discussion: A brief discussion of the model's integration into clinical practice (e.g., computational requirements, integration with hospital information systems) could be provided.
Response 6: We have expanded the clinical integration discussion in the Discussion section (paragraph 7, page 13) beyond computational requirements to address system integration pathways. The enhanced discussion covers HIS integration strategies, digital pathology platform compatibility, open-source deployment considerations, and workflow integration that complements existing diagnostic frameworks.

Comments 7: Supplementary Figure S2 can be used in the discussion section with clear reference within the text.
Response 7: We have enhanced the Supplementary Figure S2 reference in the Discussion section (paragraph 3, page 11) for improved clarity. The reference now explicitly directs readers to visual examples: "After reviewing these misclassified patches, we identified several plausible causes (see Supplementary Figure S2 for examples)."

Comments 8: The text is generally well-written. Only minor grammatical corrections (subject-verb agreement, etc.) are suggested.
Response 8:  We have conducted comprehensive grammatical review throughout the manuscript, with particular attention to subject-verb agreement and other linguistic clarity improvements. The manuscript has been thoroughly proofread to ensure professional academic writing standards.

Reviewer 3 Report

Comments and Suggestions for Authors

The article is part of a research area at the intersection between medicine and computer science that aims to develop AI applications in pathology.

In tumor pathology, breast cancer is one of the topics of significant interest in relation to the automated quantification of HER2. Automated quantification of HER2 membrane staining provides HER2+ scores, thereby reducing subjectivity and potential interpretation errors by human operators.

Between 2020 and 2025, PubMed indexed over 300 articles using AI and HER2+ as keywords. HER2 status is critical for predicting both prognosis and treatment response. The primary goals of AI-based applications are to optimize the differentiation of HER2-low and HER2-ultralow cases and to improve overall diagnostic accuracy and reproducibility.

Using WSI, the authors propose to develop and compare two transform-based models for HER2 scoring of breast cancer. Transformer models are deep learning neural networks specialized for tracking relationships in sequential data.

Therefore, the study’s topic is of interest due to its potential to contribute to the advancement of artificial intelligence technology for classifying HER2 status assessed by immunohistochemistry in breast cancer.

Introduction

The authors provide a comprehensive overview of the topic, highlighting the diagnostic and prognostic significance of HER2, technical issues that may impact reproducibility, and detailing the key advancements in automatic HER2 assessment (including AI tools).

Material and methods

The authors lay out a detailed and comprehensive presentation of the methodology. The overview of the HER2 scoring workflow, illustrated in Figure 1, makes the process easy to understand, even to readers who are unfamiliar with specific computer science and computer vision terminology. The technical details included in the method's description support the scientific validity of the proposed algorithm, effectively translating the subjective expertise of pathologists into objective, computer-based image analysis. 

However, the method's weakness lies in its small sample size (n = 66 cases), which limit the generalization of the results.

Results

The results are clearly presented and supported by appropriate iconography.

The text's section-based structure strengthens the connection between the stated objectives and the results obtained.

Discussion

The authors analyze the strengths of the two developed models in comparison to other AI solutions for HER2 scaling, as well as the study's limitations.

The text is well written, denoting - again - the solid collaboration between the pathologist who interprets the results and the computer science specialist responsible for the technical support for their automated delivery.

Recommendations

While recognizing the scientific value of the article, we believe its impact on current practice could be enhanced by addressing the following issues (as minor revision):

How could sensitivity for identifying 2+ cases be improved?

Why do 1+ and 2+ scores remain problematic for AI, despite its high overall performance?

Is there a risk of the model learning unrealistic features (coloring artefacts, etc.)?

Could models suggest manual re-evaluation of ambiguous cases?

What are the costs of the hardware infrastructure required to implement these models in pathology laboratories?

Author Response

Comments 1: How could sensitivity for identifying 2+ cases be improved?
Response 1: We have added comprehensive strategies for 2+ sensitivity improvement in the Discussion section (page 12, paragraph 5). These include: (1) ensemble approaches combining both models' complementary strengths, (2) confidence scoring mechanisms to flag borderline cases, and (3) incorporation of additional morphological features for contextual information.

Comments 2: Why do 1+ and 2+ scores remain problematic for AI, despite its high overall performance?
Response 2: We have provided comprehensive mechanistic explanation in the Discussion section (new paragraph 6, page 12). The analysis addresses fundamental limitations at both patch-level and WSI-level, including patch resolution constraints for membranous completeness assessment and WSI-level aggregation challenges where heterogeneous patch populations complicate tier-based classification, establishing the rationale for continuous scoring approaches.

Comments 3: Is there a risk of the model learning unrealistic features (coloring artifacts, etc.)?
Response 3: We have enhanced the Methods section (Data Preparation subsection, page 5-6) with artifact prevention measures. Our approach includes: (1) epithelial masking in the first stage to exclude non-tumor regions and eliminate most tissue-processing artifacts, and (2) systematic labeling of artifact-containing patches as "non-tumor" during training to enhance model robustness against staining artifact.

Comments 4: Could models suggest manual re-evaluation of ambiguous cases?
Response 4:  We have clarified our hybrid clinical integration approach in the Discussion section (paragraph 5, page 12). Our workflow provides pathologists with comprehensive interpretive information including patch-level percentages, HER2 heatmaps, and predicted scores rather than classifications alone. For ambiguous cases, pathologists can leverage the detailed heatmap visualizations and patch proportion data to make informed clinical decisions, as explicitly stated: "For such borderline cases, the manual review of HER2 heatmaps and verification of 2+ patch proportions were still necessary, highlighting the role of human-in-the-loop decision-making."

Comments 5: What are the costs of the hardware infrastructure required to implement these models in pathology laboratories?
Response 5: We have added implementation cost analysis in the Discussion section (paragraph 8, page 13), providing concrete hardware requirements and cost estimates based on our RTX4060 Ti testing configuration, with upgrade pathways for high-throughput requirements.

Round 2

Reviewer 1 Report

Comments and Suggestions for Authors
  1. How do you adapt the models? Any customization? Comment please.
  2. What about the prospect of deploying other lightweight models? Other heavy models like Densenet etc.? I asked previously about the reason for not using other deep neural network-based approach in the comparison. It is not addressed properly.
  3. The Authors should have provided a document showing the comments and giving the response below as well as mentioning wherein the text, the responses were incorporated.
